

# Temporal bright light at low frequency retards lens-induced myopia in guinea pigs

Baodi Deng[1,*], Wentao Li[2,*], Ziping Chen[3], Junwen Zeng[1] and Feng Zhao[1]

[1] State Key Laboratory of Ophthalmology, Zhongshan Ophthalmic Center, Sun Yat-sen University, Guangdong Provincial Key Laboratory of Ophthalmology and Visual Science, Guangzhou, China
[2] Huizhou Third People's Hospital, Guangzhou Medical University, Huizhou, China
[3] Guangdong Light Visual Health Research Institute, Guangzhou, China
* These authors contributed equally to this work.

Corresponding author
Feng Zhao,
zoc-zhaofeng@foxmail.com

## ABSTRACT

**Purpose:** Bright light conditions are supposed to curb eye growth in animals with experimental myopia. Here we investigated the effects of temporal bright light at very low frequencies exposures on lens-induced myopia (LIM) progression.
**Methods:** Myopia was induced by application of −6.00 D lenses over the right eye of guinea pigs. They were randomly divided into four groups based on exposure to different lighting conditions: constant low illumination (CLI; 300 lux), constant high illumination (CHI; 8,000 lux), very low frequency light (vLFL; 300/8,000 lux, 10 min/c), and low frequency light (LFL; 300/8,000 lux, 20 s/c). Refraction and ocular dimensions were measured per week. Changes in ocular dimensions and refractions were analyzed by paired t-tests, and differences among the groups were analyzed by one-way ANOVA.
**Results:** Significant myopic shifts in refractive error were induced in lens-treated eyes compared with contralateral eyes in all groups after 3 weeks (all $P < 0.05$). Both CHI and LFL conditions exhibited a significantly less refractive shift of LIM eyes than CLI and vLFL conditions ($P < 0.05$). However, only LFL conditions showed significantly less overall myopic shift and axial elongation than CLI and vLFL conditions (both $P < 0.05$). The decrease in refractive error of both eyes correlated significantly with axial elongation in all groups ($P < 0.001$), except contralateral eyes in the CHI group ($P = 0.231$). LFL condition significantly slacked lens thickening in the contralateral eyes.
**Conclusions:** Temporal bright light at low temporal frequency (0.05 Hz) appears to effectively inhibit LIM progression. Further research is needed to determine the safety and the potential mechanism of temporal bright light in myopic progression.

## INTRODUCTION

Nowadays, it is recognized that outdoor activities can repress the incidence of myopia (*He et al., 2015*; *Wu et al., 2013*; *Zadnik & Mutti, 2019*). One factor associated with this protective outdoor effect is the difference in light intensity (*French et al., 2013*; *Lingham et al., 2020*; *Rose et al., 2008*; *Sherwin et al., 2012*). However, although bright light is

reported to prevent the development of form-deprivation myopia (FDM) in all species studied so far (*Ashby, Ohlendorf & Schaeffel, 2009*; *Chen et al., 2017*; *Lan, Feldkaemper & Schaeffel, 2014*; *Smith, Hung & Huang, 2012*), results on prevention of lens-induced myopia (LIM), which seemed to be a better model of human myopia (*French et al., 2013*), are more variable (*Ashby & Schaeffel, 2010*; *Smith et al., 2013*). In addition, *Biswas et al. (2021)* recently reported that in a lens induced hyperopia (LIH) chicken model, daily exposure to high-intensity light promotes axial shortening and hyperopia in a duration dependent manner, whereas optical refocus promotes emmetropization and slows the development of LIH. One of the more surprising discoveries on this subject was the finding that daily exposure to intermittent bright light at very low frequencies (0.01 and 0.002 Hz in chicken and 4 h/day intermittent bright light consisted 1 h of high intensity LED light delivered every 2 h in monkeys) fully suppressed FDM development (*Lan, Feldkaemper & Schaeffel, 2014*; *Ramachandran et al., 2022*). These results indicated that bright light seems to indiscriminately suppress eye growth rather than suppress myopia *per se*, whereas changes in the visual environment (optical focus and temporal stimuli) have a stronger effect in slowing myopia development.

With respect to temporal stimuli, accumulating evidence suggests that in chicks, guinea pigs, cats and mice, stroboscopic flicker effectively induces myopia at low frequencies and prevents myopic drift at high frequencies (*Cremieux et al., 1989*; *Di et al., 2013a*; *Rucker, Britton & Taylor, 2018*; *Yu et al., 2011*; *Zhi et al., 2013*). Although we still do not know what the difference between low temporal stimuli under a background with high intensity light or dim/dark light is, we do know that the dynamic light source used in previous studies is presented as a square wave, which is unnatural and will cause dazzle reflex (*Plainis, Murray & Carden, 2006*). Another limitation of the previous experimental paradigms was that in these studies, the spectral composition of artificial light is not as well-distributed as sunlight (*Li et al., 2014*), which was also suggested to be an independent factor affecting myopic progression. Therefore, it is necessary to evaluate the effect of a more natural and applicable temporal bright light source in LIM for potential therapeutic application for children's myopia.

In an attempt to develop such a dynamic light source that can be applied to humans, full-spectral light with gentle changes of light intensities was applied in this study. With a well-developed visual system (*Buttery et al., 1991*) and rapid effective response to form-deprivation and optical defocus (*Howlett & McFadden, 2006*; *Howlett & McFadden, 2009*), guinea pigs have been a popular alternative for studying myopia (*Howlett & McFadden, 2006*; *Howlett & McFadden, 2009*; *Li et al., 2014*; *Luo et al., 2017*; *Yu et al., 2021*). According to previous studies, guinea pigs are born hyperopic and undergo rapid emmetropization before 3 weeks of age, which was similar to the time course for emmetropization in early childhood of humans (*Zhou et al., 2006*). Additionally, the temporal response and its development in the guinea pig retina is identical with those of human beings (*Armitage et al., 2001*; *Racine et al., 2005*). Therefore, we consider 1-week-old guinea pigs to be the ideal choice for research on temporal effects of juvenile myopia and by doing so validate the feasibility of this newly designed light source in controlling myopia in children. To avoid possible retinal damage by high level exposure to light

(*Hunter et al., 2012*), we reduced the high light intensity to a less bright level at 8,000 lux. Although the lowest temporal frequency used in previous research to explore the sensitivity of temporal modulation is 0.25 Hz (*Swanson et al., 1987*), we choose much lower frequencies here to avoid dizziness or discomfort caused by temporal light as much as possible. Here we define the two lower frequencies as the low frequency (0.05 Hz) and very low frequency (0.0016 Hz). We believe that this study makes a novel contribution to our understanding of the influence of temporal stimuli on myopia because this study was the first, to the best of our knowledge, longitudinal evaluation of the effect of bright light at very low frequency on the development of LIM.

## MATERIALS AND METHODS

### Animals housing

In the present study, male and female 1-week-old guinea pigs (*Cavia porcellus*, English short-hair stock, tricolor strain) were obtained by the Animal Experimental Centre of Zhejiang Province, China and were provided with unconstrained food and water. Two to three guinea pigs were reared in a customized cage (28.2 cm * 38.2 cm * 28.5 cm inside), which provides independent lighting conditions from the feeding room. Wiry bottom was applied to keep the hutch dry and ventilated with the room temperature controlled to $22 \pm 2$ °C. To minimize potential confounders, cages of different groups were placed next to each other and female and male animals were separated by housing. The lamps were set to be on 12:12 light/dark cycle (turned on from 8:00 AM to 8:00 PM). This experiment was carried out in accordance with the ARVO Statement for the Use of Animals in Ophthalmic and Vision Research and was approved by the animal experimentation ethics committee of the Zhongshan Ophthalmic Center (approval number: 2020-095A). Before the experiment, the eyes were checked under a slit-lamp microscope and the animals with abnormal eyes such as microphthalmia or corneal haze would be excluded. Necessary measures were used to minimize the animals suffering during the experiment. Additionally, when we observed guinea pigs suffering noticeable weight loss (rapid onset of more than 15% of body weight), weakness or dying, we considered early termination. After the experiment, all guinea pigs were euthanized by intraperitoneal injection of excessive 1% Pentobarbital sodium (300 mg/kg) followed by cervical dislocation. Before handing over the carcass to the animal center for further disposal, we checked their breathing and heartbeat to ensure that no animals survive.

### Experimental design

Myopia was induced by application of −6.00 D lenses in the right eye of each guinea pig as described by *Li et al. (2014)*. In short, a homemade Velcro mask was glued to the face of guinea pigs. The mask has appropriate holes to expose eyes, nose, mouth and ears. Another Velcro with a round plastic frame was attached around the right eye, and the frame was glued with a negative lens (−6.00 D, PMMA, diameter 18.0 mm, optical zone: 12.0 mm, base curve: 8.0 mm). Special attention was paid to ensuring that the optical center of the lens was aligned directly in front of the center of the pupil. Those lenses were checked at least once a day to ensure that they were in the correct position and clean. If the face mask
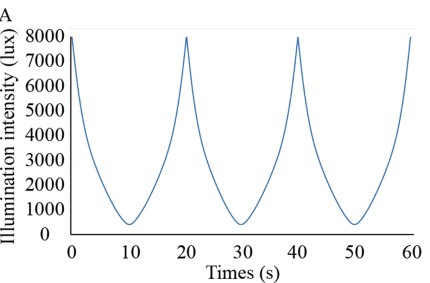
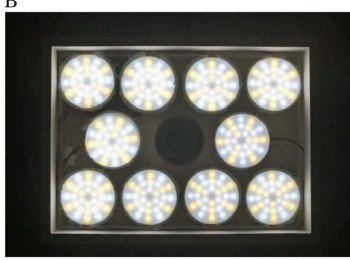

**Figure 1 Lighting conditions used in the study.** (A) Waveforms of flicker used in the experiment changed smoothly; (B) Arrangement of lamps on top of cages.

or lenses were loosened or fell off, they would be reattached at once. Additionally, once the center of the lens was found to have obvious scratches, it was replaced immediately.

A total of 82 1-week-old guinea pigs were used in this study, referring to previous related research (*Li et al., 2014*; *Luo et al., 2017*). They were marked with ear tags and the numbers on the ear tag were input into the excel table. Then RANDBETWEEN (1, 4) functions were applied on the numbers to generate the randomisation sequence. Afterwards, the numbered guinea pigs were assigned to one of the following four groups: (1) constant low illumination (CLI; $n$ = 22, 300 lux), (2) constant high illumination (CHI; $n$ = 19, 8,000 lux), (3) temporal high luminance at very low frequency (vLFL, $n$ = 21, 300/8,000 lux, 10 min/cycle), and (4) temporal high luminance at low frequency (LFL; $n$ = 20, 300/8,000 lux, 20 s/cycle).

## Lighting

Solux halogen lamps (4300k; Eiko Ltd., Shawnee, KS, USA) were used to create the dynamic simulated sunlight. The lamp was measured with a fluorospectrophotometer (HR2000; Ocean Optics, Inc., Osaka, Japan; the detection limit is 200–1,100 nm) by the Department of Physics of Sun Yat-sen University in Guangzhou, China. Except for the wavelengths between 300 and 350 nm, the spectrum emitted by this lamp effectively simulates the spectral composition of sunlight (*Li et al., 2014*). We designed the illuminance changing from low (300 lux) to high (8,000 lux) levels smoothly and automatically in a temporal wave function to achieve a single variable (Fig. 1A). To achieve the intensity of illumination and form full spectrum light needed in this study, 288 independently controlled point light sources were installed on the roof of the cage at a height of 28.5 cm from the bottom of each cage (Fig. 1B). The illumination was manipulated by a function generator (Patent No.: US201916257198) linked to the lamp. Function generators were placed on the outside of the cages. According to previous studies, the temporal sensitivity function (TSF) in guinea pigs is band-pass at bright stimulus intensities (*Armitage et al., 2001*). Additionally, in guinea pigs, the lifetime of rod desensitization is of the order of 10 s and that of the bleached pigment is presumably 10 min for recovery from saturation (*Demontis, Bisti & Cervetto, 1993*). Consequently, we chose 0.05 and 0.0016 Hz as the temporal frequencies to make the effective temporal bright light as comfortable as possible.

## Measurement of ocular parameters

Ocular parameters including refractive error and axial dimensions (anterior chamber depth (ACD), lens thickness (LT), vitreous chamber depth (VCD), and axial length (AL)) were measured before the experiment and once per week during the treatment.

Refractions were measured by handheld streak retinoscopy (66 Vision-Tech Co., Ltd., Suzhou, Jiangsu Province, China) with cycloplegia. 0.5% proparacaine hydrochloride (Alcaine; Alcon, Fort Worth, TX, USA) was dropped topically into the conjunctival sac of guinea pigs at first, followed by drops of 0.5% tropicamide and 0.5% phenylephrine (Mydrin-P; Santen, Osaka, Japan) every other 5 min for five times to induce cycloplegia. Results from the two independent skilled optometrists from Zhongshan Ophthalmic Center, who were blinding with regard to the treatment, were averaged. Refractive error was taken as the mean value of the refractive errors with the vertical and horizontal meridians of three repeated measurements and expressed as spherical equivalent (SE).

Ocular biometry was performed by A-scan ultrasonography with a probe of 10 MHz (KN-1800; Kangning Medical Device Co., Ltd., Wuxi, Jiangsu Province, China) as described by *Li et al. (2014)*. In order to achieve local anesthesia while using biepharostat, 0.5% proparacaine hydrochloride (Alcaine, Alcon) eye drops were applied to eyes before measurement. The ultrasonic probe was in direct contact with the corneal apex and carefully made sure that it was vertical to the corneal surface. The mean of the 10 repeated measurements was used for ocular parameters analysis. Due to the fact that the 10 MHz ultrasound probe does not allow choroidal measurements, the AL *in vivo* was described as the axial distance from the anterior corneal surface to the vitreo-retinal interface (*Di et al., 2013b*).

## Statistics

Data were represented as mean ± SD and statistical analyses were performed with GraphPad Prism (v7.0) (GraphPad Software Inc., La Jolla, CA, USA). Before analyzing the data, complete the normal distribution test. Paired t-tests were used to compare the relative changes between deprived eyes and non-deprived eyes within a group. Comparisons among groups were assessed by one-way ANOVA followed by Tukey's multiple comparisons test or Kruskal-Wallis test. If significant differences were detected, *post hoc* range tests were performed using the Duncan test using SPSS 25 (SPSS, Chicago, IL, USA). Statistical tests were two-tailed, and *p*-value < 0.05 was considered statistically significant.

## RESULTS

Ocular parameters of all guinea pigs at different time points are listed in Table 1. No significant difference was found in the parameters between the left and right eyes of the individual animals within any group prior to the treatment (all *P* > 0.05, see Table 1).

### Refractive errors

Although all lens-treated eyes became significantly less hyperopic than the contralateral eyes in all light conditions after 3 weeks of light exposure (Table 1), the magnitude of the response differed among the lighting conditions.

**Table 1 Biometric results (mean ± SD) of ocular parameters and changes at different time points.**

| Paradigms | Groups | Time points | Refractive error, D | ACD, mm | LT, mm | VCD, mm | AL, mm |
|---|---|---|---|---|---|---|---|
| Without lenses | CLI | Baseline | 3.60 ± 0.58 | 1.10 ± 0.05 | 2.49 ± 0.14 | 3.46 ± 0.16 | 7.24 ± 0.14 |
| | | First week | 3.39 ± 0.57 | 1.11 ± 0.05 | 2.53 ± 0.14 | 3.52 ± 0.15 | 7.33 ± 0.14 |
| | | Second week | 3.28 ± 0.54 | 1.13 ± 0.05 | 2.57 ± 0.14 | 3.57 ± 0.16 | 7.45 ± 0.14 |
| | | Third week | 3.09 ± 0.55 | 1.14 ± 0.05 | 2.58 ± 0.14 | 3.61 ± 0.16 | 7.55 ± 0.16 |
| | | Change | −0.51 ± 0.05 | 0.05 ± 0.02 | 0.10 ± 0.03 | 0.14 ± 0.05 | 0.30 ± 0.08 |
| | CHI | Baseline | 3.64 ± 0.62 | 1.09 ± 0.06 | 2.47 ± 0.14 | 3.43 ± 0.20 | 7.25 ± 0.13 |
| | | First week | 3.50 ± 0.65 | 1.10 ± 0.06 | 2.51 ± 0.15 | 3.48 ± 0.18 | 7.34 ± 0.16 |
| | | Second week | 3.39 ± 0.65 | 1.12 ± 0.06 | 2.53 ± 0.15 | 3.53 ± 0.17 | 7.40 ± 0.17 |
| | | Third week | 3.24 ± 0.62 | 1.13±0.06 | 2.55 ± 0.15 | 3.57 ± 0.18 | 7.49 ± 0.17 |
| | | Change | −0.40 ± 0.23 | 0.05 ± 0.01 | 0.08 ± 0.03 | 0.14 ± 0.05 | 0.24 ± 0.15 |
| | vLFL | Baseline | 3.95 ± 0.67 | 1.11±0.05 | 2.51 ± 0.14 | 3.44 ± 0.19 | 7.26 ± 0.09 |
| | | First week | 3.84 ± 0.79 | 1.13±0.06 | 2.55 ± 0.13 | 3.49 ± 0.20 | 7.36 ± 0.08 |
| | | Second week | 3.56 ± 0.60 | 1.15 ± 0.06 | 2.59 ± 0.13 | 3.55 ± 0.20 | 7.48 ± 0.08 |
| | | Third week | 3.36 ± 0.60 | 1.16 ± 0.06 | 2.61 ± 0.12 | 3.59 ± 0.20 | 7.53 ± 0.08 |
| | | Change | −0.59 ± 0.29 | 0.05 ± 0.03 | 0.10 ± 0.04 | 0.14 ± 0.05 | 0.27 ± 0.07 |
| | LFL | Baseline | 3.76 ± 0.62 | 1.11 ± 0.05 | 2.50 ± 0.12 | 3.44 ± 0.20 | 7.28 ± 0.09 |
| | | First week | 3.69 ± 0.54 | 1.12 ± 0.06 | 2.51 ± 0.13 | 3.48 ± 0.20 | 7.38 ± 0.10 |
| | | Second week | 3.41 ± 0.63 | 1.14 ± 0.06 | 2.55 ± 0.12 | 3.53 ± 0.20 | 7.50 ± 0.12 |
| | | Third week | 3.25 ± 0.60 | 1.15 ± 0.06 | 2.56 ± 0.12 | 3.57 ± 0.20 | 7.60 ± 0.12 |
| | | Change | −0.51 ± 0.16 | 0.04 ± 0.02 | 0.07 ± 0.03 | 0.13 ± 0.04 | 0.31 ± 0.08 |
| With -6D lenses | CLI | Baseline | 3.58 ± 0.41 | 1.09 ± 0.04 | 2.49 ± 0.14 | 3.43 ± 0.16 | 7.22 ± 0.1 |
| | | First week | 2.89 ± 0.62 | 1.1 ± 0.04 | 2.62 ± 0.13 | 3.58 ± 0.17 | 7.61 ± 0.1 |
| | | Second week | 1.78 ± 0.52 | 1.11 ± 0.05 | 2.7 ± 0.13 | 3.73 ± 0.21 | 7.74 ± 0.1 |
| | | Third week | 1.27 ± 0.48 | 1.09 ± 0.05 | 2.83 ± 0.16 | 4.07 ± 0.2 | 7.99 ± 0.09 |
| | | Change | −2.31 ± 0.64 | −0.01±0.07 | 0.35 ± 0.24 | 0.64 ± 0.27 | 0.78 ± 0.13 |
| | CHI | Baseline | 3.54 ± 0.53 | 1.08 ± 0.05 | 2.45 ± 0.14 | 3.39 ± 0.18 | 7.22 ± 0.09 |
| | | First week | 2.96 ± 0.39 | 1.1 ± 0.04 | 2.61 ± 0.14 | 3.54 ± 0.25 | 7.65 ± 0.15 |
| | | Second week | 2.55 ± 0.77 | 1.09 ± 0.05 | 2.71 ± 0.12 | 3.56v0.19 | 7.76 ± 0.14 |
| | | Third week | 1.93 ± 0.9 | 1.11 ± 0.06 | 2.85 ± 0.17 | 3.88 ± 0.24 | 7.94 ± 0.1 |
| | | Change | −1.61 ± 0.94 | 0.02 ± 0.07 | 0.4 ± 0.2 | 0.48 ± 0.25 | 0.71 ± 0.12 |
| | vLFL | Baseline | 3.67 ± 0.58 | 1.09 ± 0.05 | 2.48 ± 0.14 | 3.42 ± 0.19 | 7.19 ± 0.07 |
| | | First week | 2.8 ± 0.67 | 1.11 ± 0.04 | 2.61 ± 0.13 | 3.56 ± 0.17 | 7.58 ± 0.09 |
| | | Second week | 1.77 ± 0.58 | 1.11 ± 0.05 | 2.69 ± 0.14 | 3.64 ± 0.22 | 7.74 ± 0.1 |
| | | Third week | 1.25 ± 0.47 | 1.09 ± 0.05 | 2.83 ± 0.16 | 3.97 ± 0.18 | 8 ± 0.08 |
| | | Change | −2.42 ± 0.69 | 0 ± 0.07 | 0.35 ± 0.24 | 0.55 ± 0.3 | 0.8 ± 0.11 |
| | LFL | Baseline | 3.55 ± 0.52 | 1.11 ± 0.05 | 2.48 ± 0.14 | 3.43 ± 0.15 | 7.22 ± 0.08 |
| | | First week | 2.96 ± 0.38 | 1.09 ± 0.05 | 2.6 ± 0.13 | 3.56 ± 0.25 | 7.65 ± 0.14 |
| | | Second week | 2.68 ± 0.76 | 1.09 ± 0.05 | 2.71 ± 0.13 | 3.65 ± 0.17 | 7.76 ± 0.13 |
| | | Third week | 2.15 ± 0.89 | 1.1 ± 0.04 | 2.84 ± 0.17 | 3.87 ± 0.22 | 7.91 ± 0.09 |
| | | Change | −1.4 ± 0.94 | −0.01 ± 0.06 | 0.36 ± 0.25 | 0.44 ± 0.24 | 0.69 ± 0.12 |

**Note:**

CLI, constant low illumination; CHI, constant high illumination; vLFL, very low frequency cycles of dynamic light; LFL, low frequency cycles of dynamic light; ACD, anterior chamber depth; LT, lens thickness; VCD, vitreous chamber depth; AL, axial length. Data are presented as mean ± SD.

To directly compare the effect of light conditions on LIM, the overall refractive changes (change in the lens-treated eye ($\Delta X$) subtract change in the contralateral eye ($\Delta N$), $\Delta X$-$\Delta N$) of the animals were compared. As shown in Fig. 2A and Table 1, at the end of the experiment, refractive error in vLFL had the greatest myopic shift of $-1.83 \pm 0.66$ D (95% confidence interval CI [$-2.137$ to $-1.522$]; $n = 21$), followed by CLI ($-1.80 \pm 0.65$ D; 95% CI [$-2.09$ to $-1.5$]; $n = 22$), CHI ($-1.20 \pm 0.96$ D; 95% CI [$-1.681$ to $-0.733$]; $n = 19$), and LFL ($-0.89 \pm 0.95$ D; 95% CI [$-1.345$ to $-0.432$]; $n = 20$) (one-way ANOVA, F = 6.298, $P < 0.001$). Tukey's multiple comparisons test revealed that refractive changes in LFL had significantly less myopia shift than that of CLI ($P = 0.004$) and vLFL ($P = 0.003$). *Post hoc* analysis showed that CLI and vLFL belonged to one subset ($P = 0.898$), while CHI and LFL belonged to another subset ($P = 0.224$).

Given that both eyes were affected by the lighting conditions, we further compared the changes in refraction of eyes with or without lens conditions. Consistent with the overall refractive changes, refractive error of lens-treated eyes in vLFL group had the greatest myopic shift and that in LFL group had the least at the end of the experiment (one-way ANOVA, F = 6.298, $P < 0.001$) (Fig. 2B). Comparing different light intensities, CHI exhibited a significantly lower myopic shift than CLI ($P = 0.035$). Comparing different light temporal frequencies, LFL showed significantly lower myopic shift than vLFL in lens-treated eyes ($P < 0.001$). *Post hoc* analysis also showed that lens-treated eyes in CLI and vLFL groups belonged to one subset ($P = 0.665$), while CHI and LFL belonged to another subset ($P = 0.420$). Likewise, refractive error of the contralateral eyes in vLFL had the greatest myopic shift of $-0.589 \pm 0.279$ D (95% CI [$-0.719$ to $-0.459$]; $n = 21$), followed by CLI ($-0.511 \pm 0.119$ D; 95% CI [$-0.565$ to $-0.458$]; $n = 22$), LFL ($-0.513 \pm 0.153$ D; 95% CI [$-0.586$ to $-0.439$]; $n = 20$), and CHI ($-0.401 \pm 0.224$ D; 95% CI [$-0.512$ to $-0.29$]; n = 19) (one-way ANOVA, F = 2.750, $P = 0.048$) (Fig. 2C). Specifically, the contralateral eyes in vLFL group exhibited a significantly more myopic shift than that in CHI group ($P < 0.001$).

## Ocular dimensions

All eyes elongated throughout the experiment (Table 1). As shown in Fig. 3A, after 3 weeks, relative changes of axial length in LFL had the minimum elongation of $0.38 \pm 0.12$ mm (95% CI [$0.321$–$0.44$]; $n = 20$), followed by CHI ($0.47 \pm 0.15$ mm; 95% CI [$0.395$–$0.546$]; $n = 19$), CLI ($0.47 \pm 0.16$ mm; 95% CI [$0.398$–$0.547$]; $n = 22$), and vLFL ($0.53 \pm 0.15$ mm; 95% CI [$0.464$–$0.606$]; $n = 21$) (one-way ANOVA, F = 3.488, $P = 0.02$). Nevertheless, only LFL showed a statistically significant difference in axial elongation with vLFL ($P = 0.01$). The relative changes in axial length mainly came from the axial elongation in lens-treated eyes, which showed the minimum axial elongation of $0.693 \pm 0.115$ mm (95% CI [$0.638$–$0.748$]; $n = 20$) in LFL, followed by CHI ($0.714 \pm 0.114$ mm; 95% CI [$0.657$–$0.77$]; $n = 19$), CLI ($0.776 \pm 0.123$ mm; 95% CI [$0.721$–$0.832$]; $n = 22$), and vLFL ($0.803 \pm 0.103$ mm; 95% CI [$0.755$–$0.851$]; $n = 21$) (one-way ANOVA, F = 4.001, $P = 0.01$). No statistically significant difference in axial elongation of contralateral eyes among groups was found (one-way ANOVA, F = 2.043, $P = 0.115$).

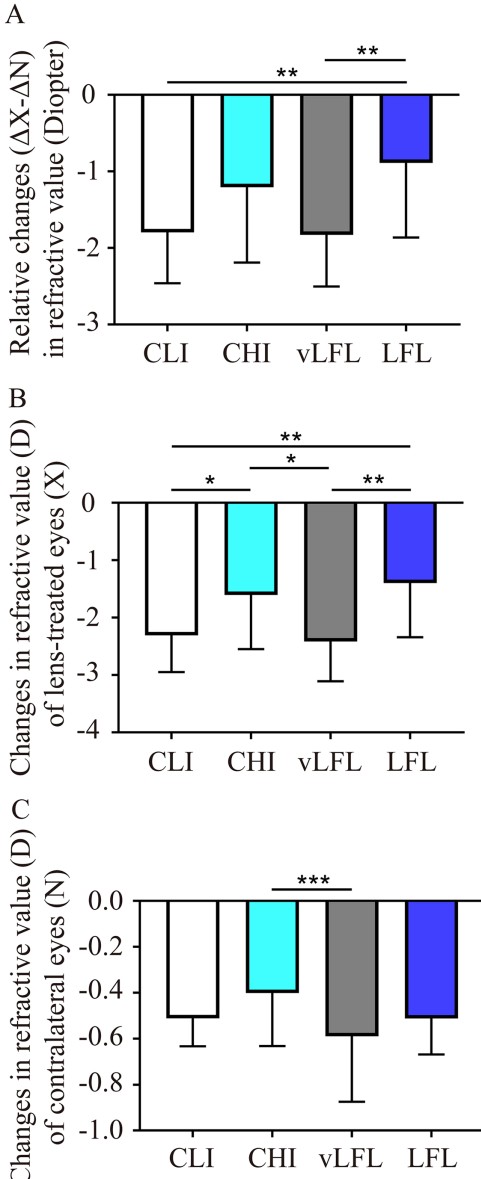

**Figure 2 Comparison of the changes of refractive error among the groups at the end of experiment.**
(A) Guinea pigs exposed to CLI and vLFL demonstrated a significant reduction in the average refractive shift (OD-OS) compared to the CHI and LFL groups. The refractive shift of lens-treated eyes (B) and the contralateral eyes (C) showed different changes. CLI, constant low illumination ($n = 22$); CHI, constant high illumination ($n = 19$); vLFL, temporal bright light at very low frequency ($n = 21$); LFL, temporal bright light at low frequency ($n = 20$); Data are presented as mean ± SD. $^*P < 0.05$, $^{**}P < 0.01$, $^{***}P < 0.001$; Error bars: ± SEM.

During the observation period, there was no obvious change of ACD in interocular difference and lens-treated eyes (one-way ANOVA, all $P > 0.05$, see Table 1). However, as shown in Fig. 3B, the ACD illustrated an increasing trend with the observation period among all groups in contralateral eyes (Friedman test, $P < 0.001$). On the contrary, compared with the initial time point, with the extension of treatment time, the relative changes of lens thickness ($\Delta X - \Delta N$) in each groups increased gradually (one-way ANOVA,

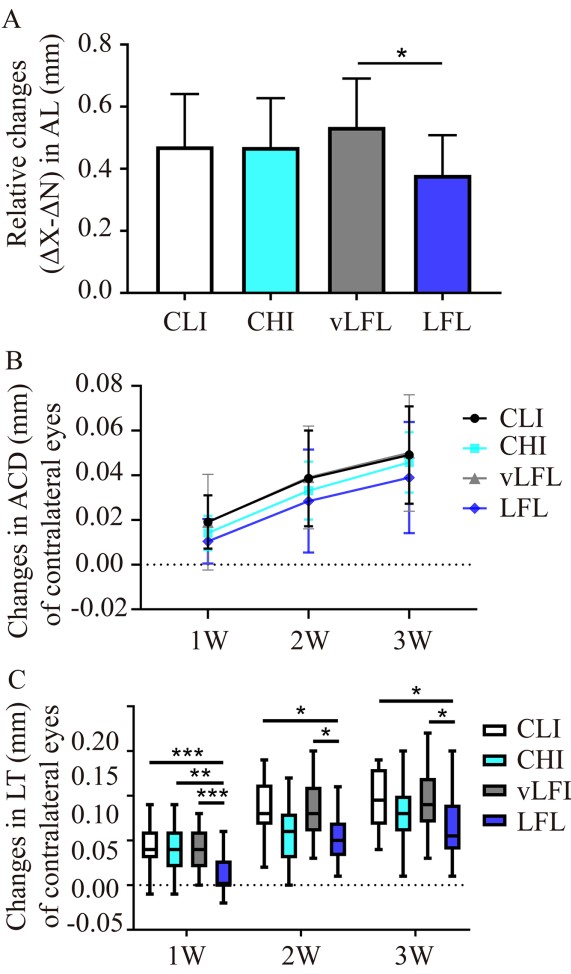

**Figure 3 The effects of four light conditions on ocular biometry.** (A) At the end of treatment, LFL exposure significantly reined in the axial elongation (OD-OS) than vLFL condition. (B) Changes in anterior chamber depth of contralateral eyes were growing with age. (C) Changes in lens thickness of contralateral eyes was less in LFL group. CLI, constant low illumination ($n = 22$); CHI, constant high illumination ($n = 19$); vLFL, temporal bright light at very low frequency ($n = 21$); LFL, temporal bright light at low frequency ($n = 20$); Data are presented as mean ± SD. *$P < 0.05$, **$P < 0.01$, ***$P < 0.001$; Error bars: ± SEM.

all $P < 0.01$), but no differences was found between the groups at any time point (all $P > 0.05$, see Table 1). This was also true for lens-treated eyes. With respect to contralateral eyes, the lens thickened less at low temporal frequencies (One-way ANOVA, F = 4.128, $P = 0.009$; $P = 0.02$ for LFL with CLI, $P = 0.022$ for LFL with vLFL) (Fig. 3C). VCD of all eyes also increased significantly with age (one-way ANOVA, all $P < 0.05$). However, changes in VCD among groups were not statistically significant (all $P > 0.05$) (Table 1).

## Correlation between changes in refractive error and ocular dimensions

Figure 4 shows the correlation between axial length elongation and refractive shift for the lens-treated eyes and contralateral eyes under each light regimen. Specifically, the decrease in refractive error (*i.e.*, more myopia) of both eyes correlated significantly with the axial length elongation (contralateral eyes: CLI: $R^2 = 0.132$; vLFL: $R^2 = 0.148$; LFL: $R^2 = 0.436$;

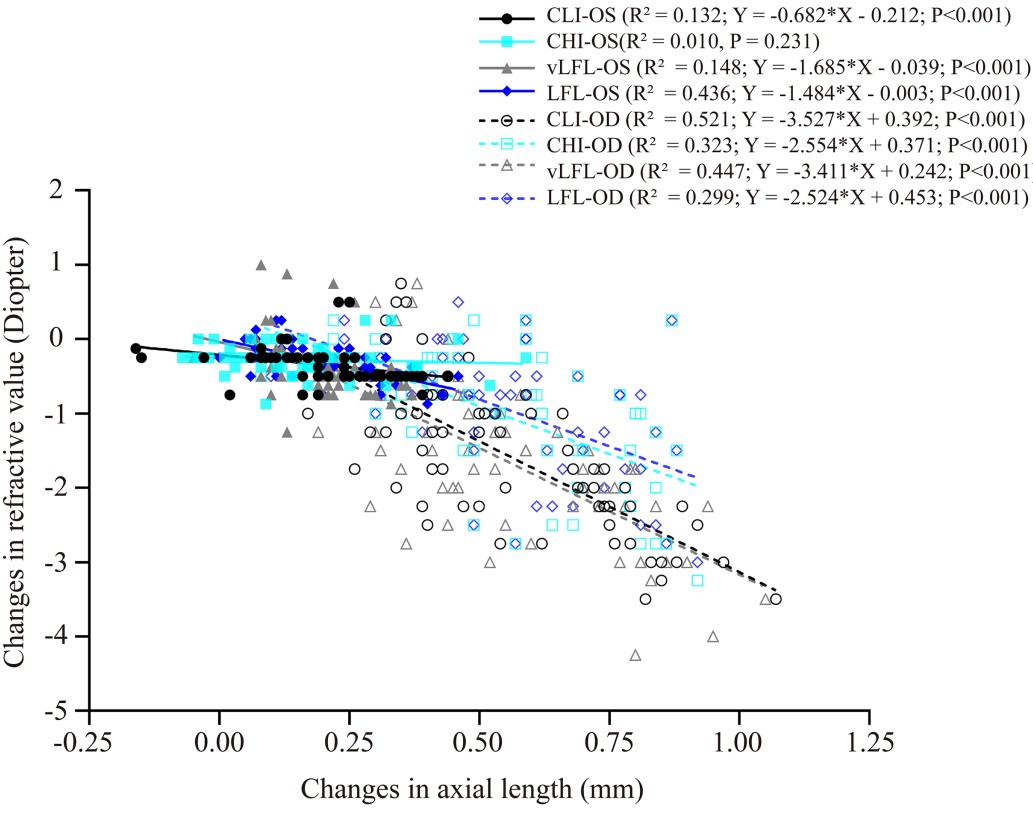

**Figure 4 Correlations between changes in axial length and refractive error.** All eyes, including the contralateral and lens-treated eyes, showed a significant correlation between changes in axial length and refractive error, except the contralateral eyes in CHI group. Solid line represented the data from the contralateral eyes (*i.e.*, OS) of all guinea pigs; long dash line represented the data from the lens-treated eyes (*i.e.*, OD) of all guinea pigs.

lens-treated eyes: CLI: $R^2$ = 0.5213; CHI: $R^2$ = 0.3226; vLFL: $R^2$ = 0.447; LFL: $R^2$ = 0.299; all $P < 0.001$), except the contralateral eyes in CHI group ($R^2$ = 0.010, P = 0.231). These results indicated that the refraction shift was largely axial origin excluding the contralateral eyes in the CHI group. To know what correlated with the refractive shift of contralateral eyes in CHI group, we analyzed the correlation of its refractive shift with other ocular parameters. However, none of the ocular parameters correlated with refractive shift of contralateral eyes in the CHI group (All $P > 0.05$; data not show).

## DISCUSSION

Here, we show that under the same housing conditions, compared with the low light conditions, the bright light conditions retard the myopic shift of LIM. The very low temporal frequency (0.0016 Hz) bright light condition produced a similar myopic shift to the low intensity illumination, while the low temporal frequency (0.05 Hz) bright light condition led to significantly less eye growth, implying a temporal sensitivity in hyperopic defocus. Additionally, LFL condition significantly slacked the thickening of lenses in contralateral eyes.

Why short outdoor time has protective effects against myopia and how myopia development is related to light parameters are two of the most studied but as yet unanswered questions in this field. Although human epidemiological studies have shown a correlation between bright light and myopia, the confounding effect of optical distance is not eliminated (*Ngo et al., 2013*). While in animal studies, myopia is indeed suppressed by bright light, which seems to be indiscriminate suppression of eye growth rather than suppressing myopia *per se* (*Biswas et al., 2021*; *Chakraborty et al., 2020*; *Chen et al., 2017*; *Feldkaemper & Schaeffel, 2013*). Consistent with these studies, bright light condition at 8,000 lux effectively retarded the decrease in spherical equivalent refraction (SER) of LIM eyes. However, no significant difference was found in the overall myopia shift.

One explanation was that bright light which was reported to be capable of retarding myopia development and enhanced hyperopic shifts of lens-induced myopia in guinea pigs was 10,000 lux (*Li et al., 2014*), while the light intensity of bright light (8,000 lux) in this study was much lower. Nevertheless, does that mean children should be exposed to continuous higher ambient light for longer periods of time? It should be noted that the refractive changes of contralateral eyes in CHI did not correlate to axial elongation. It is possible that the corneal radius of curvature was flattened under continued bright light, as was reported in chicken (*Cohen et al., 2012*; *Li et al., 1995*). Besides, recent studies suggested that a sufficient cumulative lux per day at an approximately 5,000 lux light intensity with about 2.8 h reduced 25.5 ± 4.5% myopia risk, which was equivalent to the anti-myopic effect of the same cumulative lux at a lower outdoor light intensity with much more outdoor times (*He et al., 2022*). Additionally, daily exposure to intermittent bright light at very low frequencies showed to be capable of fully suppressing FDM development (*Lan, Feldkaemper & Schaeffel, 2014*; *Ramachandran et al., 2022*). These evidences suggest that the overall photons arriving at eyes in necessary time, instead of continued bright light exposure, is imperative for myopic control. Nonetheless, temporal bright light at low frequencies showed stronger inhibitory effects on LIM also suggested a temporal sensitivity of bright light.

With respect to temporal stimuli, accumulating evidence suggests that in chickens, guinea pigs and mice, stroboscopic flicker effectively induces myopia at low frequencies and prevents myopic drift at high frequencies (*Crewther et al., 2006*; *Di et al., 2013b*; *Schwahn & Schaeffel, 1997*; *Yu et al., 2011*). Inspired by the fact that temporal stimuli are processed by midget and parasol ganglion cells in the ON and OFF pathways within the retina (*Schiller, 2010*), imbalance of ON and OFF retinal pathway activation is suggested to be the underlying mechanisms (*Crewther & Crewther, 2002*; *Crewther et al., 2006*; *Wang, Aleman & Schaeffel, 2019*). Specifically, with accumulating evidence finding that blockade of ON pathways effectively inhibited myopia progression, some studies suggested that ON pathways were the pro-myopic factor (*Crewther & Crewther, 1990*; *Crewther & Crewther, 2002*; *Crewther & Crewther, 2003*; *Smith, Fox & Duncan, 1991*). Nevertheless, other studies suggest that bright light or high frequency flicker might inhibit myopia development by stimulation of the ON pathway proposed *via* increasing dopamine (DA) pathway activation (*Chen et al., 2017*; *Chuang & Rucker, 2019*). Theoretically, the temporal bright light with a smooth decline and ascent waveform used in the current study produced a
strong stimulation of both ON and OFF pathways. Since we have not accomplished monitoring the b-wave (rapid ON response) and d-wave (slower OFF response) components of the flash ERG, we cannot be certain whether temporal bright stimuli interfered with emmetropization also *via* the imbalance of ON and OFF retinal pathway activation as it was presumed to be for stroboscopic flicker.

According to a previous study, temporal bright light with its lowest light intensity being 300 lux should activate cone-photoreceptors while rod-photoreceptors are saturated (*Joesch & Meister, 2016*). However, a recent study found that rods do saturate at beginning, but rhodopsin bleaching allows them to escape saturation at bright conditions, with the recovery time shorter at brighter background (*Kelber, 2018*; *Tikidji-Hamburyan et al., 2017*). Besides, rod activities were supposed to suppress cone flicker sensitivity and response amplitude (*Alexander & Fishman, 1986*; *Bush et al., 2019*; *Lankford et al., 2022*; *Zele, Cao & Pokorny, 2008*). Additionally, several studies have found that rod activation contributed to eye growth and myopia development (*Park et al., 2014*; *Smith, Hung & Huang, 2009*; *Smith et al., 2007*). Furthermore, rod function was supposed to be the only photoreceptor defining the dopamine release light threshold which is about 400 lux for mice (*Perez-Fernandez et al., 2019*). Combined together, we speculated that bright temporal light at a frequency whose cycle is less than rod light response period (*i.e.*, keep rod saturated) should be expected to retard myopia shift and eye growth, while the lower temporal frequencies failed to affect refractive development. The lifetime of rod desensitization in guinea pigs is supposed to be approximately 10 s, while the recovery of bleached pigment from saturation takes about 10 minutes (*Demontis, Bisti & Cervetto, 1993*). In support of our hypothesis, temporal bright light at low temporal frequencies (0.05 Hz) showed significant inhibitory effects on axial elongation and the decrease in SER than CLI and vLFL group, while the vLFL condition (0.0016 Hz) showed no effect on LIM and even promoted myopic susceptibility. However, the mentioned above evidence of rod activity under bright light condition were all from mice and there are no references from previous literature about rod function under bright condition in guinea pigs far to now. Therefore, our hypothesis needs future evidence of these temporal bright lights in mouse myopic model. Additionally, since we failed to measure if the effective intervals of repeated bright light cycles were confined to the time scale of light adaptation, we cannot be certain whether this conjecture applies to all cases. Further studies are also required to clarify the effects of bright flicker on children myopia.

The exact mechanisms underlying light effects on refractive development remain elusive. A number of hypotheses have been proposed, such as the change in depth of focus, physical activity and retinal DA activity (for reviews see. *Parry & Bowmaker, 2002*; *Perez-Fernandez et al., 2019*; *Ashby, Ohlendorf & Schaeffel, 2009*; *Ashby & Schaeffel, 2010*; *French et al., 2013*; *Muralidharan et al., 2021*; *Troilo et al., 2019*). Among them, the involvement of retinal DA seems to be most likely. In this regard, DA synthesis and release were stimulated by light and DA concentration was down regulated in experimental myopic eyes (*Boatright, Hoel & Iuvone, 1989*; *Brainard & Morgan, 1987*; *Dong & McReynolds, 1991*; *Kirsch & Wagner, 1989*; *Megaw, Morgan & Boelen, 1997*; *Rohrer, Iuvone & Stell, 1995*; *Stone et al., 1989*). The antagonists of DA receptors (DR) shown to reverse the

anti-myopic effect of bright light (*Ashby & Schaeffel, 2010*; *Chen et al., 2017*) also favored this presumption. Besides, flicker light was shown to stimulate more retinal DA release than steady light (*Kirsch & Wagner, 1989*; *Kramer, 1971*; *Umino, Lee & Dowling, 1991*). In addition, light with different spectral compositions also showed different efficiency in stimulating DA release (*Wang et al., 2018*). In particular, continuous full spectrum artificial light with no peak or valley inhibited axial elongation with higher retinal 3, 4-dihydroxyphenylacetic acid (DOPAC)/DA ratio-the metabolic efficiency of DA (*Xu et al., 2023*). Accordingly, it is reasonable for us to speculate that temporal bright light with full spectrum in the current study stimulated more retinal DA release which led to the inhibition of axial elongation. However, flicker light induced myopia in guinea pigs was corroborated by up regulating DA release (*Luo et al., 2017*). Nevertheless, a recent study in guinea pigs suggested that retinal DOPAC/DA ratio, instead of retinal DA *per se*, is associated with flicker-induced myopia (*Tian et al., 2021*). Further studies measuring levels of DOPAC/DA ratio may be helpful to better characterize the involvement of dopaminergic pathway in the temporal bright light modulation of myopia progression.

Another possible mechanism that might contribute to the anti-myopic effect of naturallight is the incremental UV exposure upregulating vitamin D in circulation (*Dixon et al., 2013*). In favor of this mechanism, accumulating evidence showed that UV exposure was inversely associated with myopia and vitamin D level was lower in myope (*Choi et al., 2014*; *Gao et al., 2021*; *Mutti & Marks, 2011*; *Tideman et al., 2016*; *Yazar et al., 2014*). It is further supported by the recent observation that calcipotriol supplement can effectively retard moues FDM (*Jiao et al., 2023*). However, several evidences suggest that low vitamin D level is not associated with myopia (*Harb & Wildsoet, 2021*; *Li et al., 2022*; *Lingham et al., 2019*; *Specht et al., 2020*; *Williams et al., 2017*). It should also be pointed out that the Solux halogen lamp applied in this study does not contain UVB ($\lambda$ = 290–315 nm) radiation, which is the only light parameter that could promote vitamin D synthesis and activation (*Chan et al., 2022*). Besides, our previous study comparing UV-free fluorescent lamps *vs* solux halogen lamps also showed no significant difference between these lamps in inhibiting LIM in guinea pigs, though with a trend of inhibiting myopia (*Li et al., 2014*). Nevertheless, it should be noted that the observed changes in refractive error are relatively small, most particularly for eyes raised at lower light levels. Other experiments have shown larger changes for low light stimuli with -6D lenses (*Wang et al., 2014*; *Wu et al., 2020*). Several studies suggested that violet light (VL, 360 to 400 nm) could effectively inhibit myopia development in experimental myopic mice and children (*Jiang et al., 2021*; *Torii et al., 2017a*; *Torii et al., 2022*; *Torii et al., 2017b*). Furthermore, wavelength-induced myopia was shown in guinea pigs as in chicks, mice, tree shrews, and human (*Gawne et al., 2017*; *Jiang et al., 2021*; *Rohrer, Schaeffel & Zrenner, 1992*; *Rucker, Britton & Taylor, 2018*; *Strickland, Landis & Pardue, 2020*; *Torii et al., 2017a*; *Torii et al., 2017b*; *Wang et al., 2018*; *Wen et al., 2022*; *Yu et al., 2021*). Since dichromatic guinea pigs also have a violet-sensitive pigment (peak at around 400 nm) (*Parry & Bowmaker, 2002*), we could not deny the possibility of the VL in solux lamps producing less compensation for optical defocus. Further studies are needed to investigate the effectiveness of the spectral distribution in

refractive development so as to test its necessity in indoor light design for children myopia inhibition.

One unexpected finding was that thickening of crystalline lenses on the contralateral eyes in the LFL group was less than that in other lighting conditions. According to previous studies, the lenticular thickness of guinea pigs increased rapidly from birth to 5 weeks of age under laboratory lighting conditions, which mainly determines the increase of the axial length (*Zhou et al., 2006*). The initial increase in lens thickness after visual experience was also reported in tree shrew (*Norton & McBrien, 1992*) and marmoset (*Graham & Judge, 1999*), which differed from children and primates whose crystalline lenses thinned throughout infancy and childhood (*Mutti et al., 2005*; *Mutti et al., 2018*; *Qiao-Grider et al., 2007*). The increase in lens thickness, assuming no change in lens curvature, would tend to increase the lens power, which was suggested to be compensated for by the steepening of the cornea, leading to the continued decline of hyperopia toward emmetropization (*Zhou et al., 2006*). In addition, the acceleration of decline in lens thickness in children is related to myopia onset and progression (*Lu et al., 2021*). Given that at the end of the experiment the guinea pigs used in this study were 4 weeks of age and lenses thickening occurred synchronously with increasing vitreous chamber depth, we considered the tardier increase in lens thickness in the LFL group as the reflection of slowing the overall eye growth. Considering that crystalline is essential to the focusing power of the vertebrate eye lens (*Roskamp et al., 2020*), how flickering light affects lenticular thickness and refractive power still needs further studies.

A limitation in the current study was that the 10 MHz ultrasound probe used in the current study does not allow choroidal measurements. According to previous studies, experimental myopia in guinea pigs shows significant changes in choroidal thickness (*Yang et al., 2023*; *Zhang et al., 2019*). The insensitive ultrasound may compromise the accuracy of axial length measurements and account for our failure to detect significant axial differences between groups. However, the results of refraction and ocular dimensions in the present study still provide support for the inhibitory effect of bright light on LIM. In this respect, since light levels affect axial responses to lenses and occluders in guinea pigs (*Li et al., 2014*; *Zhang & Qu, 2019*), in the current study, a group with the same mean illuminance as the modulated light sources might be an alternative control. However, the brightest light in the current study did not show significant influence on the axial length compared with the low light group. The impact of mean illuminance on refractive outcomes across light modulated conditions is likely to be minimal. One more shortage in the current study is that the duration of treatment is short (3 W), which might also be the reason that the axial differences between groups are not captured. Longer-term studies are needed in future studies to provide a better understanding of the prolonged effects.

## CONCLUSION

Collectively, the results of this preliminary study suggest that temporal modulation of Solux halogen lamps at low temporal frequency (0.05 Hz) could be an effective way of inhibiting LIM progression. Nonetheless, the application of these findings to humans is limited by the fact that, different from humans, guinea pigs are dichromatic and have no

fovea (*Do-Nascimento et al., 1991*; *Rohlich, van Veen & Szel, 1994*). Future studies are required to investigate how temporal stimuli affect the refractive shift and eye growth. Nevertheless, this study is helpful in understanding the effect of light environment and temporal stimuli on myopia, which may help the development of novel and effective treatment options for slowing myopia progression in children.

### Funding
This work was supported by the Science and Technology Planning Project of Guangdong Province (2016A040403016); the Science and Technology Project of Guangzhou (202102020886); and the China Educational Equipment Industry Association (CEFR20001K1). No funding or sponsorship was received for the publication of this article. The funders had no role in study design, data collection and analysis, decision to publish, or preparation of the manuscript.

### Grant Disclosures
The following grant information was disclosed by the authors:
Science and Technology Planning Project of Guangdong Province: 2016A040403016.
Science and Technology Project of Guangzhou: 202102020886.
China Educational Equipment Industry Association: CEFR20001K1.

### Competing Interests
The authors declare that they have no competing interests.

### Author Contributions
- Baodi Deng performed the experiments, analyzed the data, prepared figures and/or tables, authored or reviewed drafts of the article, and approved the final draft.
- Wentao Li analyzed the data, prepared figures and/or tables, and approved the final draft.
- Ziping Chen conceived and designed the experiments, prepared figures and/or tables, and approved the final draft.
- Junwen Zeng performed the experiments, authored or reviewed drafts of the article, and approved the final draft.
- Feng Zhao conceived and designed the experiments, analyzed the data, authored or reviewed drafts of the article, and approved the final draft.

### Animal Ethics
The following information was supplied relating to ethical approvals (*i.e.*, approving body and any reference numbers):
This experiment was carried out in accordance with the ARVO Statement for the Use of Animals in Ophthalmic and Vision Research and was approved by the animal experimentation ethics committee of the Zhongshan Ophthalmic Center.

## Data Availability

The raw data is available at Figshare: Deng, Baodi; Li, Wentao; Chen, Ziping; Zeng, Junwen; Zhao, Feng (2023). Temporal bright light at low frequency retards lens-induced myopia in guinea pigs. figshare. Journal contribution. https://doi.org/10.6084/m9.figshare.23708685.v1.

## Supplemental Information

Supplemental information for this article can be found online at http://dx.doi.org/10.7717/peerj.16425#supplemental-information.

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
