# Peer review of "Temporal bright light at low frequency retards lens-induced myopia in guinea pigs"

_PeerJ, doi:10.7717/peerj.16425_

## Round 0.1 · original submission · Major Revisions

Both reviewers have provided some valuable comments and suggestions that need to be addressed before we can proceed with the publication process.
I kindly request you to carefully review the reviewers' comments and provide a detailed response to each of their points. Please address their concerns and provide a clear explanation of how you have addressed each issue raised. Additionally, make sure to incorporate any necessary revisions and improvements based on their suggestions.

Please note that it is crucial to thoroughly address all the comments and suggestions to ensure the quality and validity of your manuscript. This will enhance the chances of its acceptance for publication in our journal.

Reviewer 1 ·

Basic reporting

The article is well written but need minor revision and more references should be added.

Experimental design

The design is sound, but more detail should be described in the study.

Validity of the findings

The validity of the findings presented in the paper is robust.

Additional comments

The authors investigated the impact of temporal bright light on lens-induced myopia (LIM) progression in guinea pigs. The study is interesting, and the manuscript demonstrates valuable insights; however, there are certain aspects that need further clarification and improvement.

1.Definition of Very Low Frequency (Line 59): The authors mention the exposure to "temporal bright light at very low frequencies." It is essential to define what is meant by "very low frequencies" to ensure clarity for the readers. This definition should be provided either within the abstract or early in the introduction.

2.Clarification of Lux Range (Line 131): The authors describe the different lighting conditions as "300~8000 lux." To enhance clarity, it is advisable to revise this expression to "300/8000 lux" in order to avoid any ambiguity regarding the range of lux values.

3.Reason for Surface Anesthesia (Line 155): The authors briefly mention surface anesthesia in guinea pigs without specifying the purpose, such as the use of a biepharostat.

4.Consistency in Terminology (Throughout): There should be consistent usage of terminology when referring to the treated eye and the control eye throughout the manuscript. This consistency is important to ensure clarity in understanding the experimental design and results.

5.Clarity of Results Description (Lines 261-277): The description of results in this section is somewhat confusing. It is recommended to simplify the explanation in this section, minimizing the use of transition words.

6.Effects of 300 Lux on Rod and Cone Activation (Line 300): An illuminance of 300 lux is not considered dim for guinea pigs. Under such conditions, how are rod cells activated? Please provide references from previous literature for clarification.

Reviewer 2 ·

Basic reporting

This manuscript investigates the effect of two VERY low temporal frequency, sinusoidal, lighting conditions (0.05 and 0.001 Hz), combined with monocular lens induced myopia, on refraction and ocular components. These conditions were compared with two constant light intensity conditions. They found that constant low illumination and low frequency cycles of dynamic light showed similar refractive development while constant high illumination and high frequency cycles of dynamic light produced less myopic refractive development. The rationale behind the choice of temporal frequency stems from a previous experiment performed by Lan et al 2014, and a novel hypothesis that rod regeneration is a controlling factor in emmetropization. The experiment and results have value, since they show effects of very LOW temporal frequencies on lens induced myopia.

Cite literature as effectively as possible to avoid unnecessary literature stacking.

In general,The manuscript is well written and the graphs are adequate.

Experimental design

1) The very low temporal frequency (0.0016 Hz) bright light condition, which had a slower temporal modulation, resulted in a myopic shift similar to that observed with low intensity illumination. On the other hand, the low temporal frequency (0.05 Hz) bright light condition, which had a faster temporal modulation, led to significantly less eye growth and therefore less myopic shift. This suggests that the temporal sensitivity of the visual system plays a role in responding to hyperopic defocus, which is an important factor in myopia development. In addition to the effects on myopia progression, the study also found that the low frequency light (LFL) condition had a significant impact on the thickening of lenses in the contralateral eyes. This implies that the beneficial effects of the LFL condition extended beyond the treated eyes and influenced the overall eye development. These findings contribute to our understanding of the role of bright light conditions and temporal frequency in regulating myopic progression. By manipulating the intensity and temporal characteristics of light exposure, it may be possible to effectively influence eye growth and potentially slow down the development of myopia. It is important to note that these findings were obtained from the specific experimental conditions and animal model used in the study. Further research is needed to validate these findings in humans and explore the underlying mechanisms responsible for the observed effects.

2)The study is particularly noteworthy because it represents the first longitudinal evaluation, to the best of the researchers' knowledge, of the influence of mesopic light (a lighting condition that combines both scotopic and photopic vision) at very low frequencies on the development of lens-induced myopia. By employing this approach, the researchers aimed to shed light on the role of temporal stimuli in myopia progression and provide valuable insights into controlling myopia development. Overall, this study contributes to our understanding of how varying light parameters, particularly in terms of intensity and temporal patterns, can potentially impact the development and progression of myopia. The author should state in the Introduction what potential downsides the model has and how they were avoided in the study.

3) The study focused specifically on lens-induced myopia, which is induced by optical defocus. It might not necessarily reflect the effects of different lighting conditions on other types of myopia, such as axial or environmental myopia. The findings should be interpreted within the context of lens-induced myopia rather than being generalized to all forms of myopia.

4) In the method, the diagram of light output describes a light source that is modulated with sharp transients. What was the logic behind using this waveform? The transients introduce higher frequencies than the ones of interest.

Validity of the findings

5) The study focused on comparing the effects of different lighting conditions, including very low-frequency light, continuous low-intensity light, continuous high-intensity light, and low-frequency light. However, the specific light parameters, such as intensity, duration, and spectral characteristics, were not discussed in detail. These parameters can significantly influence the results and should be considered in future studies.

6) Ongoing research suggests that short outdoor time and specific light parameters, such as intensity, pattern, and cumulative exposure, are important factors in understanding and potentially reducing myopia risk. Further studies are needed to unravel the exact mechanisms and establish evidence-based guidelines for optimal light exposure to prevent myopia. The author should discuss possible mechanisms in the discussion.

7) The fonts in Figures need to be uniformly sized, and there are numerous descriptions of similar results in the annotations of Figure 2.

Additional comments

8) The limitations of the study are not comprehensive. Although the current manuscript has publication value, there are still many aspects that have not been fully considered, such as The study investigated the effects of light exposure over a three-week period. While this timeframe was sufficient to observe significant refractive changes, it may not fully capture the long-term effects of different lighting conditions on myopia progression. Longer-term studies would provide a better understanding of the prolonged effects. It is important to consider these limitations when interpreting the results of the study and to conduct further research to confirm and expand upon these findings.

---

## Round 0.2 · accepted · Accept

Both reviewers are quite satisfied with your revisions and have no more questions. We look forward to sharing your work with the academic community and to your continued success in your research endeavors.

Reviewer 1 ·

Basic reporting

No further suggestions

Experimental design

No further suggestions

Validity of the findings

No further suggestions

Additional comments

No further comments

Reviewer 2 ·

Basic reporting

The author's revisions are complete, and the manuscript version is now available for publication.

Experimental design

no comment

Validity of the findings

no comment

Additional comments

no comment